# Sustainability of a school-based health intervention for prevention of non-communicable diseases in marginalised communities: protocol for a mixed-methods cohort study

Patricia Arnaiz ,[1] Larissa Adams,[2] Ivan Müller,[1] Markus Gerber ,[1] Cheryl Walter,[2] Rosa du Randt,[2] Peter Steinmann ,[3,4] Manfred Max Bergman,[5] Harald Seelig,[1] Darelle van Greunen,[6] Jürg Utzinger,[3,4] Uwe Pühse[1]

For numbered affiliations see end of article.

**Correspondence to**
Dr Patricia Arnaiz;
patricia.arnaiz@unibas.ch

## ABSTRACT

**Introduction** The prevalence of chronic, lifestyle-related diseases is increasing among adults and children from low-income and middle-income countries. Despite the effectiveness of community-based interventions to address this situation, the benefits thereof may disappear in the long term, due to a lack of maintenance, especially among disadvantaged and high-risk populations. The *KaziBantu* randomised controlled trial conducted in 2019 consisted of two school-based health interventions, *KaziKidz* and *KaziHealth*. This study will evaluate the long-term effectiveness and sustainability of these interventions in promoting positive lifestyle changes among children and educators in disadvantaged schools in Nelson Mandela Bay, South Africa, in the context of the COVID-19 pandemic.

**Methods and analysis** This study has an observational, longitudinal, mixed-methods design. It will follow up educators and children from the *KaziBantu* study. All 160 educators enrolled in *KaziHealth* will be invited to participate, while the study will focus on 361 *KaziKidz* children (aged 10–16 years) identified as having an increased risk for non-communicable diseases. Data collection will take place 1.5 and 2 years postintervention and includes quantitative and qualitative methods, such as anthropometric measurements, clinical assessments, questionnaires, interviews and focus group discussions. Analyses will encompass: prevalence of health parameters; descriptive frequencies of self-reported health behaviours and quality of life; the longitudinal association of these; extent of implementation; personal experiences with the programmes and an impact analysis based on the Reach, Efficacy, Adoption, Implementation, Maintenance framework.

**Discussion** In settings where resources are scarce, sustainable and effective prevention programmes are needed. The purpose of this protocol is to outline the design of a study to evaluate *KaziKidz* and *KaziHealth* under real-world conditions in terms of effectiveness, being long-lasting and becoming institutionalised. We hypothesise that a mixed-methods approach will increase understanding of the interventions' capacity to lead to

## STRENGTHS AND LIMITATIONS OF THIS STUDY

⇒ The cohort of children was purposely and objectively identified based on their high-risk profile; the follow-up of personal health outcomes is of additional value.
⇒ The mixed-methods approach, strengthened by the involvement of the community and key stakeholders, will enable a comprehensive evaluation of the programmes.
⇒ Internationally recognised measurement methods for quantitative, as well as qualitative, were selected and adapted to the study setting.
⇒ The COVID-19 pandemic might influence schools' decision to discontinue or adapt the post-*KaziBantu* intervention and the research team will assess, but not influence, its implementation.
⇒ The study will yield context-specific results that will be closely linked to COVID-19 measures and practices adopted by the schools.

sustainable favourable health outcomes amid challenging environments, thereby generating evidence for policy.
**Trial registration number** ISRCTN15648510

## INTRODUCTION

Socioeconomic, environmental and cultural factors exert a significant influence on children's health and well-being.[1] In impoverished communities in low-income and middle-income countries (LMICs), children are especially vulnerable to illness and lags in age-appropriate development, negatively affecting their life prospects.[2–4] This precarious situation has been aggravated by the COVID-19 pandemic, which has compromised millions of South African children's education, mental health and nutrition.[5] Furthermore, the disease profile of urban populations from LMICs is approaching

BMJ

that of high-income countries, as the proportion of deaths attributable to chronic, lifestyle-related diseases increases.[6–8] Of particular concerns are changing dietary habits and sedentary lifestyle patterns,[9 10] both known to lead to metabolic and physiological changes, such as obesity and hypertension, also in young children.[11 12] Children presenting with cardiometabolic risk factors are at particularly high risk because of their predisposition to developing non-communicable diseases (NCDs) later in life.[13 14] Thus, a reduction of undesired behaviours could potentially prevent the majority of chronic diseases, indicating a great need for prevention programmes and awareness campaigns.[15 16]

Global recommendations for physical activity (PA) stipulate a daily minimum of 60 min of moderate-to-vigorous PA for children between 5 and 17 years old. Walter showed that the levels of in-school PA among children from marginalised communities in South Africa do not meet the minimal requirements,[17] which was later confirmed by 'The Healthy Active Kids South Africa Report Card' (2018).[18] Therefore, there is an urgent need to promote PA among at-risk communities in South Africa. Several school-based lifestyle interventions have proved effective in increasing levels of PA and reducing obesity in children from different contexts.[19–21] However, very few school-based and large-scale PA intervention studies have been conducted in South Africa. The first study, *Health-Kick*, was set in disadvantaged low-income communities in Cape Town and aimed at promoting healthy eating habits and regular participation in PA in children to reduce the risk of chronic diseases.[22] Study results were not completely satisfactory and highlighted that educators play a key role in implementing lifestyle interventions in schools.[23 24] Some studies have reported beneficial effects of integrated community-based interventions involving the participation of educators and children.[25–27]

'Disease, Activity and Schoolchildren's Health' study documented the poor health status and double burden from communicable and NCDs of children in disadvantaged communities in Gqeberha, formerly known as Port Elizabeth, South Africa.[28] It also revealed the potential for improvement through PA and health literacy interventions, following the principles that PA is key for the promotion of health and well-being among school children, and that learning this at a young age will have long-lasting effects throughout life. Moreover, it emphasised the role of educators as influential and willing advocates.[29] Building on these results and experience, *KaziBantu*, a comprehensive school-based lifestyle intervention programme aiming at promoting health literacy in both children and educators of primary schools from disadvantaged communities in Nelson Mandela Bay (NMB) was developed.[30] *KaziBantu*, which is a composite Swahili and isiXhosa phrase meaning «Active People», was designed as a dual approach consisting of two school-based health promotion interventions, *KaziKidz* for children and *Kazi-Health* for educators, and was implemented as a cluster randomised controlled trial (RCT) between January and October 2019. Preliminary results show that more than half of the assessed children (544/975) presented at least one risk factor for NCDs. Educators responded positively to the intervention as they felt the programme was also focused on their health and well-being (individualised results were communicated to the study participants) and took cognizance of the difficult work environment they faced daily.

Furthermore, evidence suggests that general and work-related stress increases the risk of mental ill health (burnout, anxiety, impaired quality of sleep), as well as physical disease (coronary heart disease and musculo-skeletal disease). These factors seem to contribute to an increased risk of premature death.[31] This risk was confirmed in a large representative sample of South African educators (N=21 307) working in public schools. High stress levels, lack of job satisfaction and different stress-related physical illnesses (hypertension, heart disease, stomach ulcers, mental distress, tobacco and alcohol misuse) were reported.[32]

Although evidence exists on the effectiveness of community-based lifestyle interventions in LMICs,[33] studies have also shown that positive effects tend to disappear in the long term,[34] sometimes because programmes fail to be maintained.[35 36] In addition, there is still a scarcity of evidence regarding the feasibility of educational programmes in at-risk populations in disadvantaged school settings. We hypothesise that a comprehensive approach focusing specifically on high-risk children and their educators might increase implementation feasibility and ultimately lead to sustainable favourable health outcomes in both groups.

## Objectives

The goal of this follow-up study is to determine the long-term post-RCT effectiveness, adoption and continuity of both *KaziKidz* and *KaziHealth* school-based health interventions in disadvantaged primary schools in NMB, South Africa, under real-world conditions and amid the COVID-19 pandemic. To achieve this goal, the following objectives are set:

1. To assess the effectiveness of the interventions in promoting long-lasting, positive lifestyle changes among children and educators, thereby reducing risk factors for NCDs;
2. To evaluate the postintervention sustainability concerning the continuation of the programme implementation in the schools as well as the institutionalisation within the Eastern Cape Department of Education (ECDoE) in South Africa.
3. To provide recommendations towards guidelines for a health promotion framework, as well as its adaption to new contexts that may be translated into policy and practice.

## METHODS

### Research setting

This study is conducted in the eight schools originally selected for the *KaziBantu* RCT.[30] These are quintile three primary schools in the townships and Northern areas of NMB, in the Eastern Cape province of South Africa. The quintiles of South African schools are determined through the national poverty table and are ranged on a scale from 1 (poorest) to 5 (least poor). Quintile three schools represent no-fee paying schools and their communities are adversely affected by poverty and high unemployment rates.

### Study population

Originally, in the *KaziBantu* project, both *KaziKidz* and *KaziHealth* interventions were conducted in four schools, whereas the other four participating schools were used as controls.[30] In total, 975 children from fourth to sixth grade (aged 8–16 years) were enrolled in the study; 482 children were allocated to the intervention and 493 to the control arm. At the same time, 160 educators were enrolled among all participating schools; of those, 85 took part in *KaziHealth*, whereas 75 served as waiting-list controls.

This study will follow up children and educators from the *KaziBantu* trial. While all educators who took part in *KaziHealth* have been invited to participate in this follow-up phase, a cohort of children from the *KaziKidz* intervention, who have been identified as having an increased risk for NCDs, have been included in this study. We have defined increased risk as a child presenting at least one of the following cardiometabolic risk factors: (1) overweight or obesity, (2) elevated blood pressure or hypertension, (3) pre-diabetes or diabetes, (4) borderline or dyslipidaemia; according to age appropriate, standard classifications (table 1) and based on *KaziBantu* baseline data from July 2019. Of the 975 children who participated in *KaziKidz*, 544 were classified as «high risk» according to the above-mentioned definition. Since then, 183 children have graduated from primary school resulting in 361 eligible learners. Of those, 240 have consented to participate and have been enrolled in this follow-up study at the time this protocol was published. Of the 160 educators enrolled in the *KaziHealth* program in 2019, 105 have consented to participate and have been enrolled in this study so far.

### Intervention

The *KaziKidz* intervention consists of a tailored, ready-made teaching material toolkit for grades 1–7 and addresses children's risk factors for NCDs, health behaviours and psychosocial health. On the other hand, *KaziHealth* is designed as a workplace intervention targeting educators' health and well-being through health risk assessments and personalised lifestyle coaching. The interventions have been described in detail elsewhere[30] and all teaching materials and other project resources are freely available online.[37]

**Table 1** Classification criteria for cardiometabolic risk in children

| Risk factor | Indicators and cut-off points |
|---|---|
| Adiposity[59 60] | BMI-for-age (SD) |
| Overweight | >+1 and ≤+2 |
| Obesity | >+2 |
| Blood pressure[61 62] | SBP and/or DBP (percentiles and mm Hg) |
| Elevated blood pressure | ≥90th and <95th or <90th but ≥120/80 |
| Hypertension stage 1 | ≥95th and ≤99th or ≥130/80 |
| Hypertension stage 2 | >99th or ≥140/90 |
| Glycaemia[63] | HbA1c (mmol/mol) |
| Pre-diabetes | >39 and <48 |
| Diabetes | ≥48 |
| Dyslipidaemia[64 65] | TC (mmol/l) |
| Borderline | ≥4.40 and 5.15 |
| Dyslipidaemia | ≥5.15 |

BMI, body mass index; DBP, diastolic blood pressure; HbA1c, glycated haemoglobin; SBP, systolic blood pressure; TC, total cholesterol.

After conclusion of the *KaziBantu* RCT in 2019, all eight schools were given the freedom to conduct *KaziKidz* and *KaziHealth*. Thus, all children from grades 1–7 may have participated in *KaziKidz*, regardless of health condition. However, the implementation of *KaziKidz* and *KaziHealth* has proceeded independently and without support from the project team in all schools, under so-called «real-world» conditions. Therefore, the delivery of the programmes might have been only partially implemented. For instance, some schools might have needed to adapt *KaziKidz* to comply with COVID-19-related requirements, while other schools might have not been able to, or decided not to, continue with the intervention at all.

### Study design

This study represents an observational follow-up of the *KaziBantu* RCT and has been designed as a longitudinal study to investigate the long-term effects of both lifestyle interventions, *KaziKidz* and *KaziHealth*, on health parameters and behaviours of a cohort of high-risk children, as well as all educators originally included in the RCT. The overall effectiveness and sustainability of the programmes will be assessed using a mixed-methods approach, including quantitative and qualitative data collection methods and analyses (figure 1). Recruitment has started in March 2021. Assessments of health outcomes will be performed both for children and educators at two time points: April 2021 and September 2021, that is, 1.5 and 2 years post-RCT, to evaluate the maintenance of healthy, active lifestyles and its effect on an individual level. As

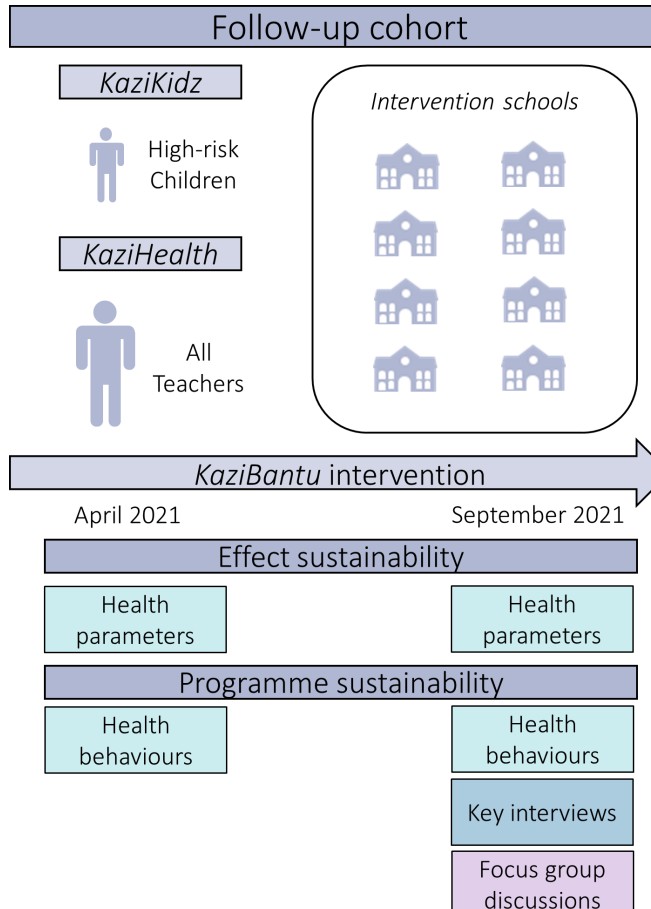

**Figure 1** Overview of the follow-up study design to assess the sustainability and effectiveness of *KaziKidz* and *KaziHealth*.

part of the postintervention evaluation, the extent of *KaziKidz*'s implementation in the schools will be obtained from school representatives at both time points to be able to correlate the observed outcomes with continued participation in the original intervention. A qualitative analysis of the school headmasters' experiences with *KaziKidz*, as well as of its incorporation into the schools' curriculum with officials from the ECDoE, will take place 2 years post-RCT, in October 2021. Additionally, qualitative analyses will also be conducted with school educators on their experiences with, and incorporation of, *KaziHealth* into their lifestyles 2 years post-RCT, in October 2021. Further information on the adoption and acceptance of *KaziKidz* by educators and parents, and of *KaziHealth* by school headmasters' and ECDoE officials will be obtained during October 2021.

This postintervention sustainability assessment does not involve the evaluation of programme delivery adequacy. Similarly, the research team will not influence the effect that «real-world» conditions might have on whether, or how, the interventions are delivered in the schools. Since the studied cohort does not involve the follow-up of a control group, participants from schools refusing the implementation would serve as comparison groups for potential health and behavioural effects.

## Data collection
Primary outcomes include anthropometric and clinical examinations, self-reported and objectively measured PA, self-reported diet and self-reported psychosocial health, while secondary outcomes include age, sex, socioeconomic status (SES) and school. Standard procedures to assess the interventions' health effects on markers for NCD risk factors will be conducted by well-trained professional staff (eg, biokineticists), who will adhere to standardised, quality-controlled protocols established during previous work of the research group in the area.

Anthropometric measurements: Body weight will be measured on a digital weighing scale and body height against a stadiometer with back erect and shoulders relaxed. Body mass index (BMI) will be calculated as weight (kg)/height (m),[2] and additionally, BMI-for-age stratified by sex will be determined for children according to WHO growth charts. Waist circumference will be measured with a steel tape between the rib cage and the iliac crest and hip circumference at the maximal circumference of the buttocks; waist-to-hip ratio will be calculated as waist circumference/hip circumference. Body fat percentage will be assessed via bioelectrical impedance analysis (BIA) with a wireless body composition monitor (Tanita MC-580; Tanita, Tokyo, Japan) in participants at least 3 hours after food consumption.

Clinical examinations: Blood pressure will be measured three times after the participant has been seating for 5 min. The Omron digital blood pressure monitor and appropriate cuffs sized 17–22 cm for children and 22–32 or 32–42 cm for adults will be used for this purpose and will be wrapped around the left arm with the bottom of the cuff placed about 4 cm above the elbow and the palm facing up. The blood pressure will be estimated as the average value of the second and third measurement. A full blood lipid profile will assess total cholesterol, low-density lipoprotein cholesterol, high-density lipoprotein cholesterol (HDL-C), triglycerides, non-HDL cholesterol and cholesterol to HDL ratio with the Alere Afinion AS 100 Analyzer point-of-care testing instrument. Glycated haemoglobin (HbA1c) will be measured from the second of two blood drops extracted from the fingertip previously cleaned with alcohol and pricked with a safety lancet, using the Alere Afinion AS 100 Analyzer. Haemoglobin concentration will be assessed for educators with the HemoCue Hb 301 system.

Objective measurement of physical behaviours: PA will be measured for each participant using an accelerometer (ActiGraph wGT3X-BT, Pensacola, Florida, USA) which will be worn around the hip, at all times, for 7 days (five weekdays and two weekend days), except for activities involving water contact.

Questionnaires: Information on PA during school hours and in their free time will be self-reported by children using adjusted questions from the PA Questionnaire for Children.[38] Dietary behaviours and nutrition will be self-reported by children through a tailored, validated food frequency questionnaire[39 40] and by educators via

a 24-hour dietary recall questionnaire.[41] The validated KIDSCREEN-10 questionnaire, European generic health-related quality-of-life questionnaire (10-domain version), will be implemented to determine children's physical and psychological well-being.[42 43] Where necessary, questionnaires will be translated into the home language of the children, isiXhosa or Afrikaans, and additionally explained orally by students of the Nelson Mandela University. Self-reported psychosocial health in educators will be assessed through general perceived stress (four-item Perceived Stress Scale),[44] work-related stress (Effort-Reward Imbalance),[45] work and family conflict (Work and Family Conflict Scale),[46] and burn-out symptoms (Shirom-Melamed Burnout Measure[47]) questionnaires. Information about educators' work attendance will be obtained from the schools' attendance registers. A self-developed questionnaire will be used to assess the continuity and extent of *KaziKidz*'s implementation in the schools and will be answered by the school representative.

The support for, and integration into the school curriculum of, *KaziKidz* and *KaziHealth* sustainability will be studied by qualitative research, employing credibility, transferability, dependability, confirmability and authenticity as quality criteria when employing qualitative methods .[48]

Interviews: Information on the strengths, challenges and prospects of the *KaziKidz* toolkit implementation in the schools and the Eastern Cape school curricula will be collected via semistructured interviews. Interviewees will include principals from all participating schools (n=8), as well as representatives from the ECDoE (n=1), Provincial Teacher Development Institute (n=1) and the Nelson Mandela University (n=1). The interview schedule pertains to the perceived strengths, weaknesses, acceptance and degree of integration of the intervention programme into the school curriculum.

Focus groups: Information on community feasibility and community acceptance of *KaziKidz* and *KaziHealth* will be gathered during 16 focus group discussion (FGD) sessions of eight participants each. For *KaziKidz*, six focus groups will be organised and will include 24 parents and 24 educators from previous intervention and control schools. The sessions will last approximately 60 min. Parents and educators will be part of different groups to focus on the feasibility and acceptance of the programme at home and in schools, respectively. For *KaziHealth*, eight focus groups will include educators from all schools, whereas two other sessions will be organised with all eight schools' principals, on the one hand, and officials from the ECDoE, on the other.

## Data management

Throughout the study, personal data will be anonymised. All data will be saved on SWITCHdrive, a Swiss non-commercial cloud data storage service for institutions of higher education. Stored data will be backed up weekly and the principal investigator will supervise the content to ensure data quality. Data collected will be accessible only to authorised personnel (investigators and ethics committee members) and exclusively used for scientific research. Quantitative data will be double-entered and validated in EpiData V.3.1 and once cleaned, transferred to STATA V.15.1 for statistical analysis. Interviews and FGDs will be audiorecorded, transcribed and coded. At the end of the study, data will be anonymised and shared using the publicly accessible data repository, Zenodo.

## Data analysis

Anthropometric, clinical indicators and objectively measured PA will be characterised by means and SD, if they are close to the normal distribution, and otherwise by medians and IQRs. Data on self-reported health status, PA and dietary patterns collected in questionnaires will be treated as categorical data and described by their frequency distributions. The interview and FGD data will be analysed using qualitative content analysis,[49] and each interview and focus group transcript will be cross-coded to locations and thematic fit. To improve intercoder reliability, two independent coders will be employed. The results of the qualitative analyses will include a taxonomy of the perceived strengths, challenges and implementation of the intervention programme.

### Cross-sectional analysis

Mixed linear models: Associations of continuous health and PA outcomes with personal characteristics (eg, sex, SES) will be assessed postintervention, that is, from data collected after the conclusion of the trial. It will be adjusted for the type of school (ie, intervention vs control) and potential clustering within schools and classes, respective random intercepts will be used.

Mixed logistic regression: For categorical outcome measures, after suitable dichotomisation of polytomous outcomes, associations with the same predictor variables as for the mixed linear models will be analysed postintervention using the same random intercepts.

### Longitudinal analysis

Mixed linear regression: Associations between longitudinal changes in continuous variables (eg, BMI) and individual characteristics, like sex and SES, will be assessed with random intercepts for schools and classes. Longitudinal changes in these outcomes will also be compared between the previous intervention and control schools after adjustment for personal characteristics. Moreover, changes in health parameters will be regressed against changes in PA and nutritional parameters. All analyses will be conducted, with and without adjustment, for the postintervention values of the respective outcomes. Also, potential associations between adjusted school-specific means of longitudinal changes in health parameters and the retrospectively assessed degrees of implementation of intervention measures in the respective schools will be assessed.

Mixed logistic regression: Longitudinal changes in categorical outcomes will be analysed by dichotomising

polytomous outcomes. Analyses will be stratified according to the postintervention value of the dichotomised outcome, using mixed logistic regression models of the follow-up outcome with the same random effects structure and the same predictor variables, as previously described, for the mixed linear models. Additional analyses will be conducted by treating the postintervention and the two consequent FU outcomes as repeated measures.

To adjust for biases associated with missing data and/or lost to follow-up, inverse probability weighting will be used. Data imputation will be considered as an additional option if the drop-out rate is moderate ( ie, <20%).

### Impact analysis

For the second objective of assessing the overall community impact of the intervention, its acceptability, adoption, implementation and maintenance will be explored at the individual, community and organisational level. For this purpose, a systematic analysis will be performed based on the RE-AIM evaluation framework and its proposed dimensions ((1) Reach, (2) Efficacy, (3) Adoption, (4) Implementation and (5) Maintenance).[50] Each dimension will be analysed separately, using pieces of data collected by the above-mentioned methods and as described in table 2, and results will be expressed as percentages. Ultimately, the combination of all five dimensions will result in a score that Glasgow and colleagues refer to as «public health impact» or «population-based effect».[50]

### Participant and public involvement

Adult participants were involved in the conception of the research question by identifying their pivotal role in both the implementation and maintenance of the intervention. They will collaborate in the methodology via interviews and FGDs, where preliminary results will be presented for their verification. Participants will also contribute to the results by reporting on the burden of the interventions and specific local needs.

Participants have been informed on the study, alongside their right to voluntary participation and withdrawal at any time without justification. Before taking part in any

**Table 2** Impact analysis evaluation based on the RE-AIM framework

| Dimensions | Social level | Evaluation criteria |
|---|---|---|
| Reach<br>Proportion of the target population that has participated in the intervention | Individual:<br>Children<br>Educators | Number and characteristics of children who:<br>► Have received at least half the planned lessons, as reported by the educators.<br>Number and characteristics of educators who:<br>► Have received the health risk assessment.<br>► Have attended at least one face-to-face coaching session. |
| Efficacy<br>Success rate of the intervention in reducing risk factors | Individual:<br>Children<br>Educators | Changes in:<br>► Anthropometric measurements.<br>► Clinical parameters.<br>► Quality of life. |
| Adoption<br>Proportion of settings that has incorporated the intervention | Organisational:<br>School educators<br>School principals | Number and characteristics of:<br>► Schools that implemented the intervention.<br>► Health-promoting activities per school. |
| | Community:<br>School educators<br>Parents | Number and characteristics of:<br>► Educators who have adopted *KaziHealth* recommendations.<br>► Parents aware of and supporting *KaziKidz* activities. |
| Implementation<br>Extent to which the intervention has been embedded in the schools under naturalistic conditions | Organisational:<br>School principals | Number of:<br>► Grades and classes participating in the intervention.<br>► Terms per school year in which the intervention is being taught.<br>► Staff involved in delivering the intervention. |
| Maintenance<br>Adherence to and dissemination of the intervention | Individual:<br>Children<br>Educators | Changes in:<br>► Physical activity and dietary behaviour. |
| | Organisational:<br>Eastern Cape Department of Education<br>Nelson Mandela University | Number and characteristics of:<br>► Schools/districts that have the intervention integrated into the curriculum/environment.<br>► Actions taken towards its permanent integration (policy), for example, teaching agenda of future educators. |

RE-AIM, Reach, Efficacy, Adoption, Implementation, Maintenance.

assessment, as proposed by this study, oral assent has been sought from children and written consent from their guardians, as well as from the participating educators.

Study results will be summarised in manuscripts and submitted for publication to open-access, peer-reviewed scientific journals. Results will also be presented at relevant national and international conferences and workshops.

## Methodological limitations

The implementation of *KaziKidz* and *KaziHealth* will not be monitored by the research team and schools were given the freedom to decide whether, and to what extent, they will deliver the health interventions. Therefore, any participating school may decide not to implement the interventions after completion of the *KaziBantu* trial. The extent of *KaziKidz*'s implementation will be assessed via questionnaire with school representatives and information about the reasons for discontinuation will be sought; this information will be considered for the feasibility assessment. Similarly, the status and extent, including reasons for discontinuation, of *KaziHealth*'s implementation will also be explored and evaluated via FGDs with school educators, headmasters' and ECDoE officials. Furthermore, for those cases where a further implementation of the interventions has not taken place, it will be aimed at measuring the corresponding children and educators at the foreseen follow-up time points.

## Discussion

Globally, a total of 41 million deaths are attributable to non-communicable, chronic conditions, such as diabetes, coronary heart diseases or cancer every year. Of those deaths, it is estimated that around 80% are preventable because they are associated with poor diets, physical inactivity, alcohol addiction, tobacco use and polluted environments. However, NCD aetiology is very complex and individuals cannot be held responsible because they live in a system that encourages unhealthy behaviours. Hence, aiming merely to change behaviours might be complicated and ineffective over time. Instead, the focus should be on tailoring interventions aimed at creating environments that promote healthy lifestyles, making the system more beneficial, accessible and affordable for everyone. This is especially important in the early stages of life, where healthy behaviours can be adopted naturally and maintained throughout life. Furthermore, NCDs are a major threat to individual and national economic development: unhealthy habits acquired during childhood will most likely lead to unhealthy adults who are unable to develop to their full potential.[51]

The COVID-19 pandemic has highlighted the increased risk for severe disease and mortality faced by people living with chronic conditions.[52] At the same time, it has jeopardised the livelihood of millions of families, disproportionately hitting the most vulnerable and taking a special toll on children.[53] By December 2020, South African children lost up to a third of the school year, thus, not being able to complete the entire curriculum and leaving many gaps in children's education. Moreover, socioeconomic inequality is also likely to have further increased, since poorer learners and schools were least able to catch up.[54]

On reopening in both 2020 and 2021, schools in South Africa have been forced to adopt strict measures to further limit the spread of COVID-19. In addition to the staggered return to the school system, the 1.5 m social distancing requirement has led to schools incorporating what they referred to as a block teaching system, whereby schools operate at 50% or less of their capacity. More information on COVID-19-related school lockdown and closure in South Africa and the Gqeberha region can be found in online supplemental material. Furthermore, the ECDoE advised schools not to exclude any of the subjects from the curriculum, but to prioritise the more heavily weighted academic subjects, such as English, Afrikaans, Mathematics and Science. Physical education is currently not a stand-alone subject in the South African curriculum but forms part of the Life Orientation/Life Skills subject. Consequently, the likelihood of physical education receiving limited attention, and even being neglected, throughout the COVID-19 pandemic is high. Hence, we believe that the implementation of the *KaziKidz* material might have been affected too, which justifies further exploration. The complexity of lifestyle interventions, such as *KaziKidz* and *KaziHealth,* and their successful implementation reside in their multidimensional nature, that is, the involvement of individual, environmental and policy factors that interact at multiple social levels (individual, community and organisational institutional). The RE-AIM framework will be followed to predict the feasibility of both interventions in terms of effectiveness, addressing whole populations, being long-lasting and becoming institutionalised.[50 55 56] The information obtained from linking the intervention integration level in the schools with the personal adoption of behavioural changes and the achieved health effects both in children and educators, will yield unprecedented results on lifestyle interventions' potential to effectively promote the acquisition and maintenance of healthy lifestyles during childhood and adulthood that will ultimately support overall community health and well-being. Furthermore, we believe that continuing with the assessments indicated in the methodology sections will provide valuable insights into the children's health amidst the COVID-19 pandemic, and facilitate contextually valid recommendations for future interventions, policy and practice.

Finally, due to the multifactorial nature of NCDs, collaboration among different sectors, such as governments, communities, academia and the private sector, is necessary. The comprehensive evaluation of a tailored school-based health promotion intervention presented in this paper could provide unique evidence and support the dissemination of the intervention and collaboration between partners from different sectors,[57] ultimately influencing local policy.[58]

**Author affiliations**
[1]Department of Sport, Exercise and Health, University of Basel, Basel, Switzerland
[2]Department of Human Movement Science, Nelson Mandela Metropolitan University, Gqeberha, South Africa
[3]Swiss Tropical and Public Health Institute, Basel, Switzerland
[4]University of Basel, Basel, Switzerland
[5]Department of Social Sciences, University of Basel, Basel, Switzerland
[6]Centre for Community Technologies, Nelson Mandela Metropolitan University, Gqeberha, South Africa

**Acknowledgements** The research team would like to thank Dr Bruce Damons (Director of Centre for the Community School, Nelson Mandela University, Gqeberha) for acting as an advisor on the project and being a key intermediary between the universities and the schools. We would also like to thank PD Dr Christian Schindler, who provided input for the statistical analysis of the study. This study takes place under the auspices of the UNESCO Chair on *'Physical Activity and Health in Educational Settings'*.

**Contributors** PA, LA and IM conceptualised the study and led the development of the general project. MG, UP, CW, RdR, PS, DvG and JU contributed to the development of the study design and methodology. MMB and HS provided statistical input and developed the draft data analysis plan. PA and LA drafted this manuscript, PA elaborated the KaziKidz's part, while LA contributed with the KaziHealth's fraction. All coauthors commented on revised drafts and approved the final version of the manuscript.

**Funding** This study is funded by the Swiss National Science Foundation (SNF; Bern, Switzerland; project no. 320030_192651).

**Competing interests** All authors have completed the ICMJE uniform disclosure form at www.icmje.org/coi_disclosure.pdf and declare: PA and UP received financial support from the SNF for the submitted work; no financial relationships with any organisations that might have an interest in the submitted work in the previous 3 years; no other relationships or activities that could appear to have influenced the submitted work.

**Patient consent for publication** Not applicable.

**Ethics approval** The proposed research will be carried out according to the international ethical, scientific and clinical practice standards established in the Declaration of Helsinki, gathered in the International Conference on Harmonisation (ICH) Guidelines, and encouraged by WHO. Prior to study conduct, ethical approval was granted by the Nelson Mandela University Research Ethics Committee (ref. no. H20-HEA-HMS-001; 21/07/2020), the Ethics Committee Northwest and Central Switzerland (EKNZ; ref. no. Req-2020–00430; 12/05/2020) and the Eastern Cape Department of Education (07/12/2020).

**Provenance and peer review** Not commissioned; externally peer reviewed.

**Data availability statement** No data are available. Individual de-identified participant data will be shared using the publicly accessible data repository Zenodo. Data will become available at the end of the study for at least 10 years.

**Author note** Joint first authorship: PA and LA contributed equally to this manuscript.

**ORCID iDs**
Patricia Arnaiz http://orcid.org/0000-0001-5626-3510
Markus Gerber http://orcid.org/0000-0001-6140-8948
Peter Steinmann http://orcid.org/0000-0003-4800-3019

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
