## [Reviewer comments · BMJ Open]

ARTICLE DETAILS

TITLE (PROVISIONAL)	Sustainability of a school-based health intervention for prevention of non-communicable diseases in marginalised communities: protocol for a mixed methods cohort study
AUTHORS	Arnaiz, Patricia; Adams, Larissa; Müller, Ivan; Gerber, Markus; Walter, Cheryl; du Randt, Rosa; Steinmann, Peter; Bergman, Manfred; Seelig, Harald; van Greunen, Darelle; Utzinger, Jürg; Pühse, Uwe

VERSION 1 – REVIEW

REVIEWER	DeVilliers, Anniza South African Medical Research Council
REVIEW RETURNED	07-Feb-2021

GENERAL COMMENTS	Dear authors. I had to read the manuscript several times before I really understood the study design. It might be because in the methods section in the abstract it is not clear that the 361 children were selected from the original study that was concluded in October 2019, i.e just before the end of the 2019 school year. The implications of this follow up study in view of the huge implications COVID-19 had on the 2020 school calendar and that will most like happen again in 2021 is not fully addressed in the protocol. Will the follow-up intervention on this at risk group take place once schools return to normality or is it already ongoing and how is the intervention adapted for these changed circumstances. We know that COVID had devastating socio-economic implications for many families especially in your study area. This could have a major influence on your outcomes and should be addressed in your protocol. It is not clear from the protocol if the selected children were identified using the full battery described in the clinical examination section and whether these measurements will be repeated (or have been repeated) at baseline. The study deals specifically with at risk adults and children and it is not clear from the protocol how the original intervention will be adapted to focus on the at-risk groups. The description of the qualitative methodology also lacks specific information on how the assessments will manage to focus on the at risk groups. To conclude: The protocol as it is at the moment is vague on whether the intervention has indeed already started and how COVID-19's impact on the school environment will be factored into the intervention. It is also vague on how, what I presume was a school-wide intervention, would or was adapted to focus on only at risk children. The qualitative methodology is also poorly described.
---

REVIEWER	Fredriksen, Maren Høgskolen Kristiania, Health science
REVIEW RETURNED	09-Feb-2021

GENERAL COMMENTS	Minor comments Multiple pages Typo: «programs» not «programmes». Typo: “program” not “programme”. Page 8 of 19, line 33 (...)body mass index (BMI) will be calculated as weight/height² Will you use the child adjusted BMI scale (isoBMI) for the children?
---

VERSION 1 – AUTHOR RESPONSE

Reviewer: 1

Dear authors. I had to read the manuscript several times before I really understood the study design. It might be because in the methods section in the abstract it is not clear that the 361 children were selected from the original study that was concluded in October 2019, i.e. just before the end of the 2019 school year.

Response: Thank you for pointing out the scarcity of background information provided in the abstract. We have modified both the abstract (lines 59 – 61) and the methods section (line 263) to enhance clarity on sample selection, for both teachers and children. Please note the consistent use of the new term “post-intervention” throughout the manuscript, introduced to reflect the follow-up nature of the present study from the previous KaziBantu study finalized in 2019.

The implications of this follow up study in view of the huge implications COVID-19 had on the 2020 school calendar and that will most like happen again in 2021 is not fully addressed in the protocol. Will the follow-up intervention on this at risk group take place once schools return to normality or is it already ongoing and how is the intervention adapted for these changed circumstances. We know that COVID had devastating socio-economic implications for many families especially in your study area. This could have a major influence on your outcomes and should be addressed in your protocol.

Response: Thank you for this indeed very useful comment. We agree that the implications of COVID-19 on our study in South Africa should be addressed in the protocol. When we first submitted the protocol, it was challenging to foresee how the situation in the schools was going to unfold. At the present moment, we are in a position where it is possible to provide much more detailed and accurate information about the course of the COVID-19 pandemic (and consequences for the schools) in the study protocol. In line with your recommendation, we have thoroughly reflected on how our results might be influenced by the current pandemic and introduced these considerations throughout the manuscript (lines 57, 118 – 121, 194, 299 – 301, 583 – 605 and 618 – 621). We hope that the revised version provides now a well-rounded evaluation of the implications that the current COVID-19 situation has on our study.

Furthermore, we would like to clarify one aspect of the study design, which we have also tried to make more clear in the manuscript. This is an observational study that will follow-up both original interventions (KaziKidz and KaziHealth) to evaluate their implementation under “real-world conditions” and amid the COVID-19 pandemic. Thus, we do not intend to adapt the interventions neither to the COVID-19 situation nor to the high-risk children, nor can we influence their implementation. The schools will single-handedly decide whether and how they will implement the interventions. Moreover, the assessment of the continuity of the intervention in the schools is a central part of the assessment. To highlight this notion, the following changes have been made:

1. Objectives have been more clearly defined (lines 189 – 194 and 199 – 201);
2. The intervention (lines 295 – 299) and study design (lines 329 – 331) have been more thoroughly described to stress the fact that the interventions will not be adapted in any way and that the study team will not interfere with their implementation;
3. More information on the liberty of schools and teachers to decide whether, and if so, to what extent, they continue implementing both interventions has been included in the limitations section (lines 538 – 540);
4. The fact that the extent to what the interventions have been implemented is part of the analysis has been stated in the abstract (line 69);
5. We have described how we intend to assess the extent of the programs’ implementation in the study design (lines 314 – 317 and 321), the data collection section (lines 393 – 395), and the limitation section (lines 542 – 547).

It is not clear from the protocol if the selected children were identified using the full battery described in the clinical examination section and whether these measurements will be repeated (or have been repeated) at baseline.

Response: Thank you for pointing out this shortcoming. The selected children were classified not using the full test battery but the indicators specified in Table 1, i.e. BMI-for-age, systolic and diastolic blood pressure, HbA1c and total cholesterol. We have now specified in the manuscript that this test battery was imported from the KaziBantu baseline assessment (lines 272 – 273). The clinical examination measurements detailed in the data collection section will be performed at timepoint 1 and 2, i.e. April and September 2021 (lines 311 – 312).

The study deals specifically with at risk adults and children and it is not clear from the protocol how the original intervention will be adapted to focus on the at-risk groups. The description of the qualitative methodology also lacks specific information on how the assessments will manage to focus on the at risk groups.

Response: Thank you also for pointing out these shortcomings. We hope that we could satisfactorily address these points now with our response to your second question: Neither the intervention will be adapted to at-risk populations nor will the qualitative assessment. The latter focuses on the experiences of different stakeholders with the interventions as a whole. We have specified this in lines 70 and 317 – 324.

Furthermore, we would like to clarify that only at-risk children will be enrolled in the study, all teachers from the previous study have been invited to participate regardless of their health status. As we understand this distinction might be confusing, we have tried to explain more clearly the sample selection in the abstract (lines 61 – 62) and methods section (lines 263 – 267) and therefore also adapted the title of the manuscript, which is now “Sustainability of a school-based health intervention for prevention of non-communicable diseases in marginalised communities: protocol for a mixed methods cohort study”.

To conclude: The protocol as it is at the moment is vague on whether the intervention has indeed already started and how COVID-19's impact on the school environment will be factored into the

intervention. It is also vague on how, what I presume was a school-wide intervention, would or was adapted to focus on only at risk children.

Response: Thank you for pointing out this issue, which is linked to the previously answered points, namely that whether the interventions are taking place is part of the study assessment, and that the intervention will not focus only on at-risk children. We hope that we were able to clarify how COVID-19 may have influence the interventions so far, and which implications COVID-19 might have on our future study progress.

The qualitative methodology is also poorly described.

Response: Thank you. We have described the qualitative methodology in more detail (lines 397 – 427) as well as provided more information regarding the analysis of the qualitative data (lines 446 – 454)

Reviewer: 2

Comments to the Author

Dear Authors, I have had the pleasure of reading this manuscript. This is an interesting protocol, and I just have a few minor suggestions for change. Please take a look at the included PDF-file.

Response: Thank you for your appreciatory comments.

Multiple pages

Typo: «programs» not «programmes».

Typo: “program” not “programme”.

Response: Thank you. We have corrected all these typos.

Page 8 of 19, line 33

(...)body mass index (BMI) will be calculated as weight/height².

Will you use the child adjusted BMI scale (isoBMI) for the children?

Response: Thank you for pointing out this issue. To classify children as overweight or obese, we have used the BMI-for-age indicator according to World Health Organization (WHO) growth charts and standards. We have specified this in the revised protocol (lines 348 – 349 and Table 1).

VERSION 2 – REVIEW

REVIEWER	DeVilliers, Anniza South African Medical Research Council
REVIEW RETURNED	13-Jul-2021

GENERAL COMMENTS	Thank you for addressing my comments comprehensively. I have picked up on a few very minor editing issues. I have put it in as comments in the attached file – contact publisher for this file. One observation that you might consider is to either use educator or teacher right through the document. It is the norm to use the term educator in South Africa.
---

	The other is that it is indeed correct to use the word programme instead of program but I do not know what the policy of BMJ is?
--	--

VERSION 2 – AUTHOR RESPONSE

Reviewer: 1

Thank you for addressing my comments comprehensively. I have picked up on a few very minor editing issues. I have put it in as comments in the attached file.

Response: Thank you for your editing corrections. We have integrated them into the document (lines 46 and 135). Please kindly note we have not integrated the correction “interventions” for “intervention’s”, as we are referring to both interventions, KaziKidz and KaziHealth, separately. One observation that you might consider is to either use educator or teacher right through the document. It is the norm to use the term educator in South Africa.

Response: Thank you for bringing up this issue, which we were unaware of, to our attention. We have replaced the term “teacher” for “educator” throughout the manuscript for consistency.

The other is that it is indeed correct to use the word programme instead of program but I do not know what the policy of BMJ is?

Response: Thank you for this comment. In the first revision round, we changed “programme” for “program” following the instruction from Reviewer 2. However, we believe both would be acceptable according to The BMJ “house writing styles”, where it is stated that “The BMJ allows a mixture of English and American spelling, depending on the provenance and main target audience of the article”.